# HDAC Inhibition with Valproate Improves Direct Cytotoxicity of Monocytes against Mesothelioma Tumor Cells

**DOI:** 10.3390/cancers14092164

**Published:** 2022-04-26

**Authors:** Clotilde Hoyos, Alexis Fontaine, Jean-Rock Jacques, Vincent Heinen, Renaud Louis, Bernard Duysinx, Arnaud Scherpereel, Eric Wasielewski, Majeed Jamakhani, Malik Hamaidia, Luc Willems

**Affiliations:** 1Molecular and Cellular Epigenetics, Interdisciplinary Cluster for Applied Genoproteomics (GIGA) and Molecular Biology, Teaching and Research Centre (TERRA), University of Liège, 4000 Liege, Belgium; clotilde.hoyos@uliege.be (C.H.); alexis.fontaine@uliege.be (A.F.); jean-rock.jacques@uliege.be (J.-R.J.); majeed.jamakhani@uliege.be (M.J.); mhamaidia@uliege.be (M.H.); 2Department of Pneumology—Allergology, University Hospital, Sart-Tilman, 4000 Liege, Belgium; v.heinen@chuliege.be (V.H.); r.louis@chu.ulg.ac.be (R.L.); bduysinx@chuliege.be (B.D.); 3Department of Pneumology and Thoracic Oncology, CHU Lille, 59037 Lille, France; arnaud.scherpereel@chru-lille.fr (A.S.); eric.wasielewski@chru-lille.fr (E.W.)

**Keywords:** pleural mesothelioma, monocyte, macrophage, tumor microenvironment, histone deacetylase, valproate, cytotoxicity

## Abstract

**Simple Summary:**

Tumor-associated macrophages and monocyte myeloid-derived immunosuppressive cells are associated with bad prognosis in malignant pleural mesothelioma (MPM). This study shows that peripheral blood monocytes can, nevertheless, be cytotoxic for MPM tumor cells. This cytotoxic activity that involves direct cell-to-cell contact can be improved with a lysine deacetylase inhibitor (VPA), opening new prospects for further improvement of still unsatisfactory MPM therapies.

**Abstract:**

The composition of the tumor microenvironment (TME) mediates the outcome of chemo- and immunotherapies in malignant pleural mesothelioma (MPM). Tumor-associated macrophages (TAMs) and monocyte myeloid-derived immunosuppressive cells (M-MDSCs) constitute a major fraction of the TME. As central cells of the innate immune system, monocytes exert well-characterized functions of phagocytosis, cytokine production, and antibody-dependent cell-mediated cytotoxicity (ADCC). The objective of this study was to evaluate the ability of monocytes to exert a direct cytotoxicity by cell-to-cell contact with MPM cells. The experimental model is based on cocultures between human blood-derived monocytes sorted by negative selection and mesothelioma cell lines. Data show (i) that blood-derived human monocytes induce tumor cell death by direct cell-to-cell contact, (ii) that VPA is a pharmacological enhancer of this cytotoxic activity, (iii) that VPA increases monocyte migration and their aggregation with MPM cells, and (iv) that the molecular mechanisms behind VPA modulation of monocytes involve a downregulation of the membrane receptors associated with the M2 phenotype, i.e., CD163, CD206, and CD209. These conclusions, thus, broaden our understanding about the molecular mechanisms involved in immunosurveillance of the tumor microenvironment and open new prospects for further improvement of still unsatisfactory MPM therapies

## 1. Introduction

Malignant pleural mesothelioma (MPM) is a rare cancer arising from mesothelial cells in the pleura, mainly caused by occupational asbestos exposure. Due to the widespread use of asbestos, the incidence of MPM is still increasing worldwide despite complete or partial bans in most countries [1]. Overall survival post-diagnosis is very poor and depends on the histology of the tumor: epithelioid (12–27 months), biphasic (8–21 months), and sarcomatoid (7–18 months) [2].

Besides multimodal options that are restricted to a limited number of patients, antiproliferative chemotherapy based on the combination of a crosslinking agent (cisplatin or carboplatin) and an antifolate (pemetrexed) has been the standard-of-care for unresectable MPM for many years [3,4,5]. Cisplatin blocks the S phase of the cell cycle through covalently bound DNA adducts or via intrastrand crosslinks. Pemetrexed is a multifolate antagonist that impairs de novo synthesis of triphosphate deoxyribonucleotides (dNTPS) through inhibition of thymidylate synthase (TS), dihydrofolate reductase (DHFR), and glycinamide ribonucleotide formyltransferase (GARFT), thereby indirectly affecting DNA replication. This first-line chemotherapy is considered as palliative since patients are refractory to the cisplatin/pemetrexed regimen or rapidly relapse after treatment. Options for a second-line treatment include other cytotoxic drugs such as antimetabolites (gemcitabine/2′-deoxy-2′,2′-difluorocytidine), vinca alkaloids (vinorelbine/navelbine), or topoisomerase inhibitors (doxorubicin) [6]. Meta-analysis of 49 second-line clinical trials reveals a disappointing median progression-free survival of 3.4 months [6]. Recently, immunotherapy with two immune checkpoint inhibitors targeting PD-1 (nivolumab) and CTLA-4 (ipilimumab), proved efficacy by extending MPM patient’s overall survival to 18.1 months compared to 14.1 months with standard chemotherapy [7]. The efficacy of the immunotherapy was particularly impressive in the non-epithelioid subtypes of MPM. However, with an objective response rate of 40%, only a subset of patients benefits from this treatment. Currently, combinations of chemo- and immuno-therapies (i.e., cisplatin, pemetrexed, and pembrolizumab/durvalumab) are evaluated in phase II/III clinical trials (trials NCT02784171, NCT04334759). A major drawback of this approach is the additive and perhaps synergistic toxicity [8]. Alternative options of chemoimmunotherapy include immune checkpoint inhibitors in combination with low-dose chemotherapy that would transiently activate error-prone DNA damage repair (DDR) or tolerance (DDT) pathways. The idea is to generate non-synonymous mutations that would be translated into neoantigens in the presence of a fully competent antitumor immune response [9].

The main limitation impeding MPM immunotherapy is the strong immunosuppressive microenvironment promoted by tumor-associated macrophages (TAMs) and monocyte myeloid-derived immunosuppressive cells (M-MDSCs) [10]. Indeed, TAMs that massively infiltrate MPM tumors are correlated with poor prognosis [11,12,13,14,15,16]. Consistently with their role of macrophage precursors, circulating monocytes are also associated with a worse overall survival of MPM patients [17,18,19,20]. As central cells of the innate immune system, monocytes exert well-characterized functions of phagocytosis and cytokine production (i.e., IL-1β, IL-6, IL-8, IL-10, TNF-α) [21,22,23]. Furthermore, monocytes directly kill other cells via antibody-dependent cell-mediated cytotoxicity (ADCC) [24]. This process is initiated by the recognition of an antigen expressed on the target cell by specific antibodies. Upon interaction of the Fc domain of these immunoglobulins with their Fcγ receptors (FcγRs), monocytes release TNF-α and mediate cell death. Similarly to monocytes, macrophages also orchestrate innate immunity through phagocytosis and T-lymphocyte activation. Furthermore, macrophages can be directly cytotoxic for mesothelioma cells via a particular caspase-independent apoptosis process called oxeiptosis [25]. In this mechanism, reactive oxygen species (ROS) and peroxynitrites produced by macrophages activate the Kelch-like ECH-associated protein 1/nuclear factor erythroid 2-related factor 2 (Keap1/Nrf2) pathway in the target cell [26]. Live-cell imaging reveals that onset of oxeiptosis is initiated after a transient contact between the macrophage and the tumor cell, independently of phagocytosis and ADCC [25]. However, in MPM, the tumor microenvironment shapes the phenotype of infiltrating monocytes towards an immunosuppressive state which is predicted to impair macrophage-directed cytotoxicity [14,27,28].

In this context, the aims of this study are (i) to investigate the ability of blood-derived human monocytes to induce tumor cell death by direct cell-to-cell contact and (ii) to identify pharmacological approaches to enhance this cytotoxic activity.

## 2. Materials and Methods

### 2.1. Cell Lines and Primary Monocyte Cultures

Mesothelioma M14K (Cellosaurus CVCL_8102, epithelioid subtype) and ZL34 (CVCL_5906, non-epithelioid subtype) cell lines were kindly provided by Sakari Knuutila (Laboratory of Cytomolecular Genetics, Haartman institute, University of Helsinki, Finland), James Rheinwald (Dana Farber Cancer Research Institute, Boston, MA, USA), and Emanuela Felley-Bosco (Laboratory of Molecular Oncology, University Hospital Zürich, Switzerland). M14K and ZL34 were cultured in Dulbecco’s Modified Eagle Medium (DMEM, Lonza, Basel, Switzerland) supplemented with 10% of heat-inactivated fetal bovine serum (FBS, Gibco, Waltham, MA, USA) and antibiotics (penicillin–streptomycin 10,000 U/mL, Lonza, Basel, Switzerland) (referred to as complete DMEM). The THP-1 monocytic cell line (CVCL_0006, TIB-202) originating from peripheral blood of a pediatric patient with acute monocytic leukemia was cultured in Roswell Park Memorial Institute 1640 medium (RPMI-1640, Gibco, Waltham, MA, USA) supplemented with 10% of FBS (Gibco, Waltham, MA, USA) and antibiotics (penicillin-streptomycin 10,000 U/mL, Lonza, Basel, Switzerland) (Complete RPMI). The cells were maintained at 37 °C in a humified atmosphere containing 5% CO_2_. THP-1 monocytes were differentiated into macrophages with 16.2 µM of phorbol 12-myristate 13-acetate (PMA, Abcam, Cambridge, UK) for 48 h [29,30].

To isolate primary human monocytes, buffy coats were obtained from healthy donors (Red Cross of Belgium). The use of human samples was approved by the institutional ethic committee of the Liège University Hospital (Sart-Tilman) under reference #2012/8. Peripheral blood mononuclear cells (PBMCs) were isolated by density gradient centrifugation on Lymphoprep (1.077 g/mL, Stemcell Technology, Vancouver, BC, USA). PBMCs (70 × 10^6^ cells) were washed, blocked in PBS containing 10% of heat-inactivated FBS (PBS/FBS10%) before labeling with anti-CD3-PE (50X, Beckman Coulter, Brea, CA, USA), anti-CD19-PC5 (50X, Beckman Coulter, Brea, CA, USA), and anti-CD56-BV421 (50X, BD Biosciences, Franklin Lakes, NJ, USA) during 1 h at 4 °C. After a wash in PBS/FBS10%, monocytes were isolated by negative selection using a cell sorter flow cytometer (BD FACSAria III, BD Biosciences, Franklin Lakes, NJ, USA). Cells were collected, counted, and resuspended in complete RPMI supplemented with 1% of sodium pyruvate (ThermoFisher Scientific, Waltham, MA, USA), 1% of non-essential amino acid solution (MEM, ThermoFisher Scientific, Waltham, MA, USA), and 0.1% of 2-mercaptoethanol (ThermoFisher Scientific, Waltham, MA, USA).

Cell lines (M14K, ZL34, and THP-1) and primary monocytes were incubated with different concentrations of the HDAC inhibitor, valproate (VPA, Sigma Aldrich, St. Louis, MO, USA), or the EZH2 methyltransferase inhibitor (EPZ005687, EPZ, Bio-Connect, Huissen, The Netherlands).

### 2.2. Metabolic Activity

THP-1 monocytes (4 × 10^4^ cells/well in a 96-well plate) and primary monocytes (1 × 10^5^ cells/well in a 96-well plate) were incubated, respectively, during 48 and 24 h in the presence of different concentrations of VPA or EPZ. Metabolic activity was evaluated using the MTS assay (CellTiter 96 Aqueous One Solution Cell Proliferation assay, Promega, Madison, WI, USA), which is based on the conversion of the tetrazolium salt into formazan via the enzymatic activity of NAD(P)H-dependent dehydrogenase enzymes present in metabolically active cells. After addition of 20 µL MTS solution to 100 µL cell cultures, the optical density was measured at 490 nm with a Multiskan GO spectrophotometer (ThermoFisher Scientific, Waltham, MA, USA).

### 2.3. Apoptosis in Cocultures of Monocytes and Mesothelioma Cells

M14K mesothelioma cells (2 × 10^6^ cells) were labeled with 5 mM of carboxyfluorescein succinimidyl ester (CFSE, Sigma Aldrich, St. louis, MO, USA) in 1 mL of DMEM during 7 min at 37 °C in the presence of 5% CO_2_. THP-1 monocytes (1 × 10^5^ cells in 24-well plate) were untreated (mock) or treated with PMA (16.2 µM), VPA (2.5 mM), or EPZ (10 µM) for 48 h. After washing, THP-1 monocytes were mixed with CFSE-labeled M14K cells (1 × 10^4^ cells) and further cultured for 48 h. Primary monocytes (2 × 10^5^ cells in a 24-well plate) and CFSE-labeled M14K cells (10 × 10^3^ cells) were cocultured for 24 h in the presence of VPA (2 mM). Floating and adherent cells were collected, washed with PBS, and resuspended in 100 µL of annexin binding buffer (10 mM Hepes, 140 mM NaCl, 2.5 mM CaCl_2_, pH 7.4, BD Biosciences, Franklin Lakes, NJ, USA). Cells were then labeled with annexin V-APC (5 µL, Immunotools, Friesoythe, Germany) during 15 min at room temperature. After adding 150 µL of annexin binding buffer, labeled cells were recorded by flow cytometry (BD FACSCanto, BD Biosciences, Franklin Lakes, NJ, USA). CFSE and annexin V double-positive events were considered as apoptotic M14K cells.

### 2.4. Live Cell Imaging

THP-1 monocytes (3 × 10^3^ cells in a 96-well plate) were untreated (mock) or treated with PMA (16.2 µM), VPA (2.5 mM) or EPZ (10 µM) for 48 h. After washing the medium, the THP-1 cells were cocultured with CFSE-labeled M14K (3 × 10³ cells). Primary monocytes (6 × 10^4^ cells in a 96-well plate) were mixed with CFSE-labeled M14K cells (3 × 10^3^ cells). To evaluate cell death, annexin V-APC (5 µL, Immunotools, Friesoythe, Germany) or propidium iodide (0.5 µM, ThermoFisher Scientific, Waltham, MA, USA) was added to each well. The monocyte-M14K cocultures were maintained at 37 °C in a humified atmosphere containing 5% CO_2_ and recorded hourly with a Incucyte imaging S3 Live-Cell system (Sartorius, Göttingen, Germany) equipped with a 20× objective. The percentages of CFSE-positive M14K cells labeled by annexin V or propidium iodide were calculated with the Incucyte S3 software.

To assess cell motility, Incucyte images were further analyzed with the CellTracker 1.1 software (http://celltracker.website/index.html, accessed on 15 November 2021) [31]. Uneven signal intensity distributions were homogenized from the background by bicubic interpolation (vignetting correction). Moreover, paraxial imaging mismatches were adjusted by automatic alignment. Total migration and average speed were determined by manual cell tracking at each time set. Total migration and average speed were determined from 10 cells for each condition in triplicate.

Cluster formation between CFSE-labeled M14K cells and monocytes was quantified with the ImageJ software. The area of 10 delineated clusters was determined with the ROI Manager tool in triplicate.

### 2.5. Confocal Microscopy

CFSE-labeled M14K cells (5 × 10^4^ cells in a 24-well plate) were cultured on a coverslip. Freshly sorted primary monocytes (2 × 10^5^ cells) were then added to M14K cells. After 24 h, cells were washed with PBS and then with PBS containing 2% of heat-inactivated FBS (PBS/FBS2%). Cells were thereafter incubated with an anti-CD33 monoclonal antibody (1/200, Immunotools, Friesoythe, Germany) during 30 min at 4 °C, washed with PBS/FBS2%, and labeled with Alexa Fluor 647 (1/1000, Invitrogen, Waltham, MA, USA) during 30 min at 4 °C. After a wash in PBS/FBS2%, cells were resuspended in PBS and fixed with paraformaldehyde (PFA) 4% in the dark for 15 min. After fixation, cells were washed in PBS and stained with DAPI (ThermoFisher Scientific, Waltham, MA, USA) during 15 min in the dark at room temperature. Slides were mounted with ProLong Glass Antifade Mountant (ThermoFisher Scientific, Waltham, MA, USA) and scanned by confocal microscopy (Confocal super resolution Zeiss LSM880 AiryScan Elyra S1, Zeiss, Jena, Germany). Images were computed with the Imaris software (Oxford Instrument Imaris, Zurich, Switzerland).

### 2.6. H3K27me3 and Kac Immunofluorescence

For H3K27 trimethylation and histone acetylation, cells were collected, washed in PBS, fixed in PFA 4% during 15 min in the dark, and permeabilized in Triton X-100 0.5% during 15 min at room temperature. After a wash with PBS/FBS 10%, cells were incubated with rabbit anti-acetylated histone (1/400, Cell Signaling, Danvers, MA, USA) during 12 h at 4 °C or with rabbit anti-trimethyl-histone H3 lysine 27 (1/500, Merck Millipore, Burlington, MA, USA) during 1 h at room temperature. After washing in PBS/FBS 10%, cells were labeled with Alexa Fluor 488 anti-rabbit (1/1000, Invitrogen, Waltham, MA, USA) during 30 min at 4 °C. After a wash in PBS/FBS 10%, cells were labeled with DAPI (ThermoFisher Scientific, Waltham, MA, USA) in the presence of PBS, mounted on a coverslip in ProLong Glass Antifade Mountant (ThermoFisher Scientific, Waltham, MA, USA) and scanned with an epifluorescence microscope (Nikon Eclipse Te2 microscope, Nikon, Tokyo, Japan) equipped with a 20× objective. Images were analyzed with the NIS-Element AR software (Nikon, Tokyo, Japan). For flow cytometry analysis, cells were collected, washed in PBS, fixed in PFA 4% during 15 min in the dark, and permeabilized in Triton X-100 0.5% during 15 min at room temperature. After a wash with PBS/FBS 10%, cells were incubated with rabbit anti-acetylated histone (1/400, Cell Signaling, Danvers, MA, USA) during 12 h at 4 °C or with rabbit anti-trimethyl-histone H3 lysine 27 (1/500, Merck Millipore, Burlington, MA, USA) during 1 h at room temperature. After washing in PBS/FBS 10%, cells were labeled with Alexa Fluor 488 anti-rabbit (1/1000, Invitrogen, Waltham, MA, USA) during 30 min at 4 °C. After a wash in PBS/FBS 10%, cells were stained with Draq5 (Invitrogen). Labeled cells were recorded by flow cytometry (BD FACSCanto, BD Biosciences, Franklin Lakes, NJ, USA).

### 2.7. RNA Sequencing and Bioinformatic Analysis

Human blood monocytes isolated by flow cytometry (2 × 10^6^ cells) were cocultured with CFSE-labeled M14K cells (1 × 10^5^ cells in 6 well plate) in the presence (sample D1 + VPA) or absence (sample D1) of VPA (2 mM) for 24 h. Floating and adherent cells were recovered in cold PBS/EDTA 1 mM by pipetting and gentle scrapping. Monocytes were purified by negative selection based on CFSE labeling using a flow cytometer (BD FACSAria III sorter, BD Biosciences, Franklin Lakes, NJ, USA). As control, 1 × 10^6^ uncultured monocytes (sample D0) were washed in PBS and stored at −80 °C.

RNA from D0, D1, and D1 + VPA monocytes was isolated using the RNeasy Micro Kit (QIAGEN, Hilden, Germany). RNA integrity and quantity were evaluated with a bioanalyzer (Agilent Technologies, Santa Clara, CA, USA). When the RNA Integrity Number (RIN) was > 8, cDNA libraries were generated based on the SMARTer ultra-low mRNA (poly-A selection). Sequencing of the libraries (2 × 150 bp) was performed on a Novaseq 6000 system (Illumina, San Diego, CA, USA).

Quality controls including base quality, sequence duplication levels, and adapter contents of FASTQ reads were performed with FASTQ tools (version 0.11.9). Trimmomatic (version 0.39) was thereafter used to filter out low quality reads, minimum length reads, and Illumina universal adapters contamination (filtration parameters: SLIDINGWINDOW:4:20 TRAILING:3 MINLEN:36). rRNA contamination was removed with bwa mem (version 0.7.17). rRNA-free trimmed reads were mapped to the human genome (hg18, Genome Reference Consortium GRCh38) using STAR (version 2.7.9.a) and aligned reads were further marked for duplicates with MarkDuplicates (version 2.26.3). The read count table was eventually generated using Featurecounts (version 2.0.1).

Differential expression analysis between D1 + VPA vs. D0 and D1 vs. D0 conditions was performed using the R/Bioconductor DESeq2 package (version 1.32.0) without correction for patient heterogeneity. Differentially expressed genes (DEGs) (p-adj < 0.05 and |Log_2_FC| > 1) were obtained in each comparison and ranked according to adjusted *p*-values. Enriched pathway analysis of significant genes (p-adj < 0.05) against GO:MF, GO:BP, and KEGG databases were performed using the gprofiler2 package (version 0.2.1). Publicly available datasets analyzed in this study can be found on the library: https://www.ncbi.nlm.nih.gov/Traces/study/?acc=PRJNA807510&o=acc_s%3Aa, accessed on 21 February 2022.

### 2.8. Statistics

Standard statistical analyses were performed with the GraphPad Prism 8 software. Data were expressed as means ± standard error of the mean of at least three independent experiments. The normality of the distribution was evaluated based on the Shapiro–Wilk test and equality of variance was verified by the Brown–Forsythe test. Assuming normal distributions and variance equality, one-way analysis of variance (ANOVA 1) followed by Dunnet or Tukey multiple comparison tests were performed. In other conditions, means were compared with a Kruskal–Wallis test followed by a Dunn’s test of multiple comparisons. When only two populations were analyzed, means were compared with an unpaired *t*-test when distributions were normal and variances equal. If not, means were compared based on the Mann–Whitney test.

## 3. Results

### 3.1. HDAC Inhibition Promotes THP-1 Cytotoxicity towards MPM Cells

The first objective of the study was to investigate the ability of monocytes to induce tumor cell death by direct cell-to-cell contact. We used a well-characterized model based on THP-1 monocytic cell line. As control, THP-1 cells were differentiated with PMA (16.2 µM) during 48 h [29,30]. In order to make distinction between tumor cells and monocytes, M14K were labeled with CFSE and then cocultured with untreated (Mock) or PMA-differentiated THP-1 according to the protocol schematized in Figure 1A. The cocultures of M14K and mock THP-1 were recorded hourly by time-lapse Incucyte imaging, in the presence of annexin V-APC (Figure 1B and Appendix A) or propidium iodide (Figure 1C and Appendix A). Annexin V labels early apoptosis while propidium iodide reveals irreversible cell death through plasma membrane permeabilization [32]. Both fluorescent markers labeled CFSE^+^ M14K in contact with mock THP-1 monocytes, indicating onset of cell death. More importantly, the videos clearly illustrate that apoptosis occurred after a transient contact between M14K and THP-1 cells. The percentages of annexin V^+^ CFSE^+^ and PI^+^ CFSE^+^ double-positive M14K cells gradually increased with time (Figure 1D,E, respectively). Although cytotoxic interactions between M14K cells and mock THP-1 occurred, there was no statistically significant difference between the percentages of apoptotic M14K cells in the different conditions: absence of monocytes (mock, in black), presence of THP-1 (mock THP-1, in green) or PMA-differentiated THP-1 (PMA, in orange). A series of pharmacological inhibitors of tyrosine kinases and histone modulators was, therefore, evaluated at different doses and cell ratios to modulate the cytotoxicity of THP-1 monocytes (data not shown). Among these, valproate (VPA), a HDAC inhibitor previously evaluated in a clinical trial, significantly increased THP-1-mediated cytotoxicity towards M14K cells (in blue, Figure 1D,E and Appendix A) [33]. In contrast, an inhibitor of the EZH2 histone methyltransferase of the Polycomb repressive complex 2 (EPZ) had no statistically significant effect in eight independent experiments (EPZ, in grey, D,E and Appendix A). These conclusions were validated by flow cytometry analysis performed after 48 h of coculture (Figure 1F). Note that flow cytometry revealed an effect of PMA that was not observed by Incucyte imaging. This apparent discrepancy was related to the experimental conditions. Incucyte imaging and flow cytometry were performed at different cell THP-1/M14K ratios (1/1 and 10/1, respectively). High cell ratios were not compatible with Incucyte time-lapse imaging due to the resulting fluorescence saturation that led to misrepresentation of the number of events. In flow cytometry, high cell ratios were required to analyze significant numbers of events. The experiments were extended to another cell line of the non-epithelioid subtype (ZL34, annexin V, Appendix A). Epifluorescence microscopy confirmed the inhibitory effects of VPA and EPZ on lysine deacetylation and H3K27 trimethylation, respectively (Figure 1G,H). Time-lapse imaging further revealed that PMA and VPA promoted externalization of phosphatidylserine on THP-1 monocytes as indicated by their red fluorescence in the presence of annexin V-APC (Figure 1I). Notwithstanding, the phenotype of THP-1 differentiated with PMA or VPA markedly differed in terms of cell shape, motility, and membrane receptor. While PMA increased the cell size and granularity (Figure 1I and Appendix A) and reduced migration indicating monocyte-to-macrophage differentiation, VPA had no significant effect on THP-1 motility (Figure 1J–L). Moreover, VPA-treated THP-1 present a significantly higher expression of CD14 membrane receptor compared to mock and PMA-treated THP-1 (Appendix A).

Taken together, these data point out that VPA promotes THP-1 cytotoxicity towards M14K cells.

### 3.2. Blood-Derived Monocytes Exert a Direct Cytotoxic Activity against MPM Cells

Although THP-1 have been widely used to investigate monocyte activity, this model only imperfectly recapitulates the phenotype of primary cells. For example, THP-1 monocytes are devoid of the FcγRIII receptor (CD16) [34]. Therefore, blood-derived monocytes were enriched from buffy coats of healthy donors using density gradient centrifugation and subsequent triple-negative sorting by flow cytometry after labeling cells with anti-CD3, anti-CD19, and anti-CD56 antibodies (Figure 2A). This selective protocol yielded highly purified (>97.5%) and viable monocyte populations (Figure 2B). Further characterization for CD14 and CD16 receptors revealed, as expected, the different phenotypes of classical, intermediate, and non-classical monocytes (Figure 2C). In order to investigate their ability to interact and kill MPM cells, primary monocytes were cocultured with CFSE^+^ M14K cells. After labeling with an antibody directed against the sialic acid-binding Ig-like lectin 3 (Siglec-3 or CD33), which is specific to the myeloid lineage, cells were analyzed by confocal microscopy. Representative images shown in Figure 2D revealed that monocytes (CD33, in red) directly interact with M14K cells (CFSE, in green) and induce their nuclear fragmentation (DAPI, in blue), indicating the onset of apoptosis (Appendix A).

### 3.3. VPA Positively Modulates the Cytotoxic Activity of Blood-Derived Monocytes against MPM Cells

To further quantify their cytotoxic activity, primary monocytes (Mock) were cocultured with CFSE-labeled M14K cells and recorded by time-lapse imaging. Both annexin V-APC (Figure 3A and Appendix A) and PI (Figure 3B and Appendix A) fluorescent markers confirmed that primary monocytes (Mock) induce M14K cell death after a transient contact. Quantification of the Incucyte data set showed that primary monocytes were cytotoxic for M14K cells particularly in the presence of a sub-toxic dose of VPA (2 mM) (Figure 3C,D and Appendix A). These conclusions were validated by flow cytometry analysis performed at 24 h of coculture (Figure 3E) and at different monocyte-M14K ratios (Appendix A). Computing Incucyte images with the CellTracker software unveiled that VPA increased the total migration and average speed of monocytes (Figure 3F–H). Another marked effect of VPA is the induction of clusters between monocytes and M14K cells (Figure 3I). Cellular aggregation was quantified from Incucyte data by measuring the cluster area formed by M14K cells with one or several monocytes. Time course kinetics revealed that cluster area increased with time in the presence of monocytes (in green) particularly when the coculture was incubated with VPA (in blue) (Figure 3J).

These results thus demonstrate that VPA promotes the direct cytotoxicity of primary monocytes towards MPM cells.

### 3.4. VPA Primarily Affects Binding and Downregulates M2 Markers

In order to get a deeper insight into the mechanisms promoted by VPA, the transcriptomic profiles of cocultured monocytes were established by RNA sequencing. For this purpose, monocytes were sorted by flow cytometry after being cocultured with M14K cells (Figure 4A). RNAs were extracted from monocytes prior (D0) and after 24 h of coculture, in the absence (D1) or in presence of VPA (D1 + VPA). Complementary DNA (cDNA) libraries were amplified and sequenced with a Novaseq 6000 device. After adequate quality controls, reads were stripped of adaptors and aligned against the human reference genome. Read counts were calculated from the alignment data and differential gene expression analysis (DGEA) between experimental conditions was performed (Appendix A). As indicated by the volcano plot of Figure 4B, there was no significant differentially expressed gene (DEG) at p-adj < 0.05 that was associated with the coculture (D1 vs. D0). In other terms, the short-term culture of monocytes in contact with tumor cells only marginally affected their transcriptome. In contrast, VPA modified the transcriptome of monocytes (Figure 4C), consistently with the ability of this HDAC inhibitor to induce gene expression [35,36]. The heatmap of Figure 4D further showed the top list of DEGs affected by VPA during the coculture. Gene ontology classification and pathway enrichment analysis highlighted that protein binding (GO:0005515), carbohydrate binding (GO:0030246), small molecule binding (GO:0036094), exogenous protein binding (GO:0140272), antigen binding (GO:0003823), ion binding (GO:0043167), virion binding (GO:0046790), carbohydrate derivative binding (GO:0097367), hydrolase activity (GO:0016740), and chromatin binding (GO:0003682) predominantly characterized the effect of VPA (Figure 4E and Appendix A). Concomitantly, membrane receptors associated with the M2 phenotype were also statistically significantly downregulated in conditions D1 + VPA compared to D0: CD163 (Log_2_(Fold Change) −1.09; p-adj 0.05), CD206 (Log_2_(Fold Change) −1.13; p-adj 0.05), and CD209 (Log_2_(Fold Change) −1.64; p-adj 0.03) (Figure 4F).

Taken together, these data reveal that VPA primarily affects binding and downregulates expression of M2 markers by monocytes cocultured with MPM cells.

## 4. Discussion

In this study, we showed (i) that blood-derived human monocytes induce tumor cell death by direct cell-to-cell contact, (ii) that VPA is a pharmacological enhancer of this cytotoxic activity, (iii) that VPA increases monocyte migration and their aggregation with MPM cells, and (iv) that the molecular mechanisms behind VPA modulation of monocytes involve a downregulation of the membrane receptors associated with the M2 phenotype, i.e., CD163, CD206, and CD209.

The monocyte/macrophage lineage specific CD163 is the high affinity scavenger receptor for the hemoglobin–haptoglobin complex. The mannose receptor CD206, a C-type lectin notably expressed by macrophages, is involved in the resolution of inflammation. CD209 (DC-SIGN) is another C-type lectin that has a high affinity for the intercellular adhesion molecule-3 (ICAM3) and therefore functions as an adhesion molecule. By decreasing the level of expression of these three receptors, VPA may thus affect differentiation of monocytes to M2 macrophages/TAMs (Figure 4F) [37]. Although this is speculative, reduced expression of adhesion receptors may also favor the motility of monocytes as indicated by the increase in their migration and average speed (Figure 3F–H). At first glance, the reduced expression of adhesion receptors is less consistent with the ability of monocytes to form clusters with MPM cells (Figure 3I,J). It is nevertheless possible that increased kinetics of migration favors cell-to-cell interactions. Gene ontology has highlighted that binding (GO:0005488), including protein binding (GO:0005515), carbohydrate binding (GO:0030246), small molecule binding (GO:0036094), exogenous protein binding (GO:0140272), antigen binding (GO:0003823), ion binding (GO:0043167), virion binding (GO:0046790), carbohydrate derivative binding (GO:0097367), and chromatin binding (GO:0003682), are the most significant hallmarks of VPA activity in monocytes cocultured with M14K cells (Figure 4E). The remarkably pleiotropic effects of VPA include binding and destabilization of the Sodium Voltage-Gated Channel Beta Subunit 1 (SCN1B:Log_2_(Fold Change) 3.58; p-adj 0.005) (Figure 4C) [38]. By binding to SCN1B, VPA affects voltage-gated sodium channel function; thereby blocking propagation of action potentials and preventing epilepsy. VPA further acts as a pharmacological chaperone that binds to the Visual G Protein-Coupled Receptor Rhodopsin destabilizing its proper folded conformation [39]. More strikingly, the volcano plot of Figure 4C highlights that MEIS3 is by far the most upregulated gene (MEIS3: Log_2_(Fold Change) 5.10; p-adj 1.40 × 10^−6^) upon VPA treatment. MEIS3 is a transcriptional regulator that directly activates the expression of 3-Phosphoinositide Dependent Protein Kinase 1 (PDPK1) [40]. This kinase participates in activation of AKT/PKB and NF-κB pathways, regulation of Ca^2+^ uptake, and motility of vascular endothelial cells. By contrast, PDPK1 also provides negative feedback to toll-like receptor-mediated NF-κB activation in macrophages. Moreover, CD1B is the most downregulated gene upon VPA treatment (CD1B: Log_2_(Fold Change) −4.18; p-adj 5.13 × 10^−3^). CD1B belongs to the family of non-classical major histocompatibility complex class-I-like molecules and is expressed by monocytes/macrophages. The function of CD1B pertains to lipid antigen presentation at the cell surface of αβ and γδ T cells [41,42]. Whether these genes are involved in monocyte cytotoxicity requires further investigation with agonists/antagonists and genetic inactivation.

The broad biological effects of VPA on innate immunity mediated by monocytes/macrophages are complex and still misunderstood. A series of reports correlate VPA treatment to immunosuppressive functions such as reduction of inflammation, phagocytosis, adhesion, and cytokine production [43,44,45,46,47]. For example, VPA favors M2 polarization of RAW 264.7 macrophages in response to LPS by changing the cytokine profile (increase in IL-10 and decrease in IL-12p70, IL-6, and TNF-α) and the pattern of costimulatory molecules (increase in CD86 and decrease in CD80 and CD40) [48]. In LPS-stimulated THP-1 monocytes, VPA inhibits NF-κB activation and diminishes the secretion of both IL-6 and TNF-α. Furthermore, VPA also inhibits the expression of inducible nitric oxide synthase (iNOS), thereby reducing NO production in IFN-γ-treated RAW264.7 macrophages. In contrast, other evidences highlight that VPA also induces immunostimulatory effects. For example, VPA promotes acetylation of mitochondrial superoxide dismutase-2 (SOD2) and stimulates ROS production [49]. VPA also induces a significant increase in the expression of genes belonging to the monocytic differentiation network, including CD11b [48]. Furthermore, VPA activates transcription of pro-inflammatory cytokines (IL-12, IL-6, IFN-γ, and TNF-α) associated with M1 markers [50]. The pleiotropic effects of VPA on innate immunity are further complexified by the favored differentiation of the monocyte/macrophage lineage from myeloid hematopoietic progenitors [46,51,52].

In order to modulate this broad diversity of effects, the most likely hypothesis is that VPA affects the cell transcriptome via epigenetic mechanisms [53]. As a broad-spectrum HDAC inhibitor, VPA indeed inhibits deacetylation of histones and counterbalances the ionic interactions between positively charged lysines and phosphates of the DNA. As a result, the chromatin adopts an open conformation leading to increased RNA transcription and DNA replication [54]. As a short-chain fatty acid, VPA chelates the Zn^2+^ ion in the catalytic pocket of HDACs and inhibits their lysine deacetylase activity. Complex and selective transcriptional changes occur during monocyte-to-macrophage differentiation with both increased and decreased H3K27 acetylation at promoters [55]. In our study, the volcano plot of Figure 4B shows that monocytes did not significantly modify their transcriptome during the one-day culture, indicating the limited, or absence of, differentiation into macrophages. These experimental conditions, thus, differ from those set in other studies that investigated longer periods (48–72 h) of coculture and consequently monocyte differentiation in the presence of tumor cells [27,56]. Moreover, markers associated to monocyte differentiation were not significantly differentially expressed. Those markers include PU.1 (Log_2_(Fold Change) −0.07; p-adj 0.99), IRF8 (Log_2_(Fold Change) −0.25; p-adj 0.74), KLF4 (Log_2_(Fold Change) −0.42; p-adj 0.45), KLF2 (Log_2_(Fold Change) −0.33; p-adj 0.51), CEBPB (Log_2_(Fold Change) −0.26; p-adj 0.77), NR4A1 (Log_2_(Fold Change) −0.18; p-adj 0.61), MAFB (Log_2_(Fold Change) −0.38; p-adj 0.60), and Flt3 (Log_2_(Fold Change) −0.30; p-adj 0.66). In contrast, statistically significant transcriptional changes characterized the effect of VPA (Figure 4C,D). Although unsupervised gene ontology primarily unveiling binding (GO:0005488) is consistent with functions of cell clustering, other direct mechanisms of VPA cannot be formally excluded. For example, VPA also acts as a pharmacological chaperone that interacts with membrane-bound receptors and destabilizes their conformation [39].

In this report, we showed that VPA improves direct cytotoxicity of human blood-derived monocytes towards epithelioid MPM cells. The experimental model is based on highly purified monocytes sorted by triple-negative selection using flow cytometry and time-lapse imaging. The conclusions extend pioneering studies based on partially purified monocytes and poorly sensitive assays (^51^Cr release, ^3^H thymidine incorporation) [57,58,59,60]. Therefore, these technical settings did not reveal the direct cytotoxicity of unstimulated monocytes in basal conditions. It is also important to keep in mind that our experimental conditions did not include human serum or antibodies against tumor antigens, thereby excluding the onset of ADCC [24,61]. Single-cell imaging by Incucyte also highlighted the involvement of a transient contact between monocytes and M14K just prior to cell death based on annexin V and propidium iodide labeling (Appendix A). Finally, confocal microscopy validated the nuclear fragmentation of M14K cells in contact with CD33-positive monocytes sorted by CD3/CD19/CD56 triple-negative selection. Notwithstanding similar observations obtained with ZL34 non-epithelioid cells cultured with THP-1 (Appendix A), additional experiments are planned to assess cytotoxicity of primary monocytes in other mesothelioma subtypes (sarcomatoid and biphasic). Besides, further technical improvements in our approach may include culture conditions in 3D spheroid models [27,56] combined with light-sheet fluorescence microscopy [62]. Microfluidics has also recently opened prospects to simultaneously investigate single cell cloning, live cell–cell interactions, and their transcriptional profiling [63]. These technical improvements that associate phenotype and function would not only confirm the direct cytotoxicity of monocytes towards MPM cells but also provide information about their diversity, their cytotoxic mechanisms, and kinetics of expression. Of particular interest will be the focus on the newly identified monocyte subpopulation that distinctively expresses a cytotoxic gene signature (PRF1, GNLY, CTSW) [64].

Many open questions remain about the involvement of the monocyte–macrophage lineage in the context of mesothelioma. In clinical studies, the presence of immunosuppressive macrophages in MPM biopsies correlates with poor overall survival [13,65,66]. Furthermore, low lymphocyte-to-monocyte ratios are associated with poor prognosis [17,18,19,20]. Consistently, culture models reveal that monocytes differentiate into M2-type macrophages upon contact with MPM cells [14,27,28,56,67]. Pleural fluid from MPM patients promotes the immunosuppressive phenotype of macrophages through prostanoid prostaglandin E2 [14]. These immunosuppressive macrophages inhibit T-cell proliferation, decrease cytotoxicity of tumor antigen-specific CD8+ T lymphocytes, and limit chemotherapy efficiency [27,28]. Moreover, the C-C motif chemokine ligand 2 (CCL2) is a chemoattractant for monocytes that differentiate them into M2 macrophages via the M-CSF/CSF1R signaling pathway [28,56]. Pharmacological inhibitors targeting COX-2 (celecoxib) and CSF1R signaling (pexidartinib and GW2580) efficiently impair differentiation of monocytes into M2 macrophages [28,68,69]. A clinical trial currently evaluates the potential therapeutic improvement of celecoxib in mesothelioma (INFINITE NCT03710876) [4]. Alternatively, a monoclonal antibody that inhibits CSF-1 receptor (CSF-1R) activation may be further investigated (NCT01494688) [70]. Although less specific, another strategy to prevent macrophage immunosuppression would be to deplete phagocytic cells with liposome-encapsulated clodronate that efficiently reduced tumors in a mouse model of mesothelioma and that was previously evaluated in other cancers (i.e., Bonefos in NCT00873808) [71].

These different strategies that reduce monocyte infiltration or deplete phagocytes have the major drawback of interfering with their essential roles in cancer immunity. Besides antigen presentation, expression of cytokines, activation of cytotoxic T-lymphocytes, and phagocytosis of apoptotic cancer cells, our results show that macrophages [25] and monocytes (this report) are directly cytotoxic for MPM cells. This activity is modulated by different epigenetic regulators in macrophages (EZH2) and in monocytes (HDACs). The final outcome of pharmacological inhibitors targeting epigenetic modifications is even complexified by their effects on monocyte recruitment and antitumor activity [27]. It is, therefore, likely that more specific agonists/antagonists will be required to preserve the anticancer functions of monocyte/macrophages while impairing their immunosuppressive effects. In this paradigm, an essential parameter will be the toxicity of the candidate compounds. A randomized phase 3 trial (NCT00128102) assessing a monotherapy with a more potent HDAC inhibitor (i.e., suberoylanilide hydroxamic acid, Vorinostat) did not improve overall survival [72]. In contrast, VPA was well tolerated and improved a second line chemotherapy [4,33,73,74].

## 5. Conclusions

In this study, we showed (i) that blood-derived human monocytes induce tumor cell death by direct cell-to-cell contact, (ii) that VPA is a pharmacological enhancer of this cytotoxic activity, (iii) that VPA increases monocyte migration and their aggregation with MPM cells, and (iv) that the molecular mechanisms behind VPA modulation of monocytes involve a downregulation of the membrane receptors associated with the M2 phenotype i.e., CD163, CD206, and CD209. This study, thus, broadens our understanding about the molecular mechanisms involved in immunosurveillance of the tumor microenvironment and opens new prospects for further improvement of still unsatisfactory MPM therapies.

## Figures and Tables

**Figure 1 cancers-14-02164-f001:**
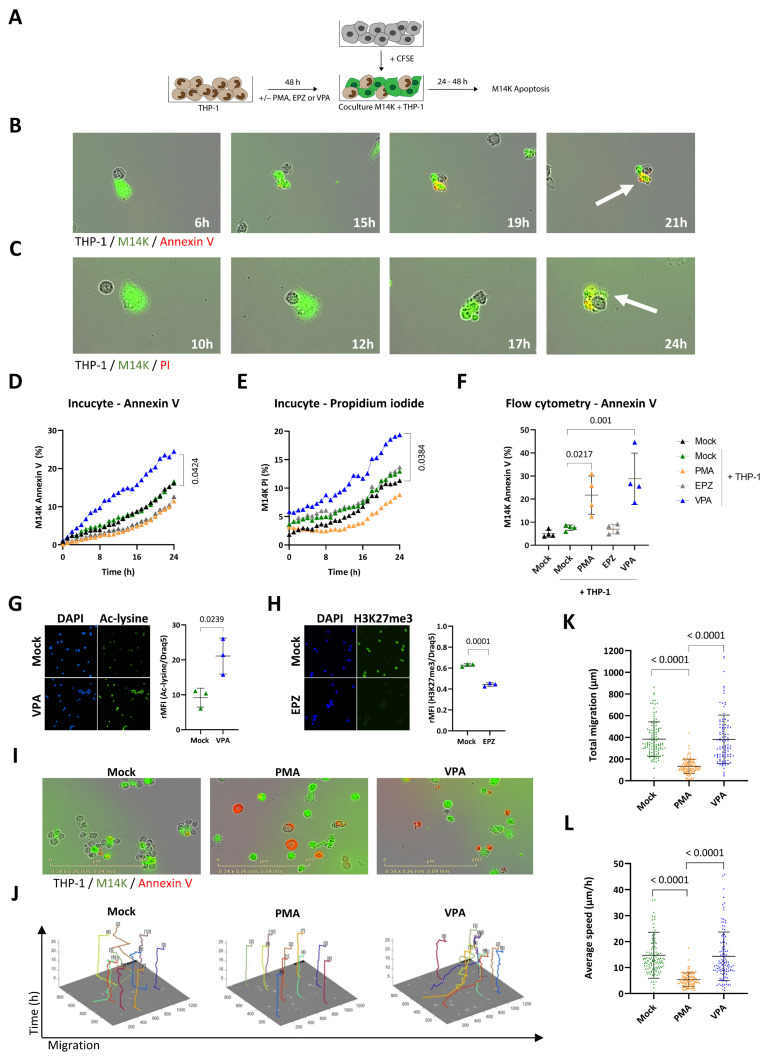
Direct cytotoxicity of THP-1 monocytes in contact with mesothelioma M14K cells. (**A**) Experimental design. THP-1 monocytes were treated with 16.2 µM of PMA, 2.5 mM of VPA, or 10 µM of EPZ005687 (EPZ) for 48 h. After washing the medium, THP-1 monocytes were cocultured with CFSE-labeled M14K cells at 1/1 (Incucyte) and 10/1 (Flow cytometry) ratios during 24 or 48 h. Apoptosis of M14K cells was evaluated by Incucyte S3 time-lapse microscopy and flow cytometry. (**B**,**C**) Untreated monocytes (Mock) and M14K cells were cocultured in the presence of annexin V-APC (**B**) or propidium iodide (**C**) and recorded by time lapse microscopy (Incucyte) for 24 h. (**D**) The percentage of annexin V positive CFSE-labeled M14K cells was calculated based on Incucyte imaging data. (**E**) The percentage of propidium iodide positive CFSE-labeled M14K was deduced from Incucyte imaging data. Data (**D**,**E**) are means of eight independent experiments. Statistical significance was evaluated using a one-way ANOVA test with Tukey’s multiple comparison test. (**F**) After 48 h of coculture, cells were labeled with annexin V-APC and analyzed by flow cytometry. Apoptosis of M14K cells (%) was determined based on the double positive CFSE and annexin V labeling. Data are expressed as means of four independent experiments ± standard deviations. Statistical significance was evaluated using a one-way ANOVA with Tukey’s multiple comparison test. (**G**,**H**) THP-1 monocytes were labeled with anti-acetylated lysine (**G**) or anti-H3K27me3 (H) primary antibodies followed by Alexa Fluor 488 conjugate. The cells were stained with DAPI and mounted on slides prior to fluorescent microscopy. The graphs of relative mean fluorescence intensities (rMFI) were obtained by flow cytometry after acetylated lysine and H3K27me3 labeling and Draq5 staining. (**I**) Representative Incucyte images are shown. (**J**) Motility of THP-1 monocytes was determined with the CellTracker software using Incucyte images. (**K**,**L**) Based on CellTracker data, total migration (**K**) and average speed (**L**) of the THP-1 monocytes were determined for 10 cells per condition in triplicate. Data are presented as the means (± standard deviations) of four independent experiments. The statistical significance was evaluated with non-parametric Kruskal–Wallis followed by a Dunn’s multiple comparison test.

**Figure 2 cancers-14-02164-f002:**
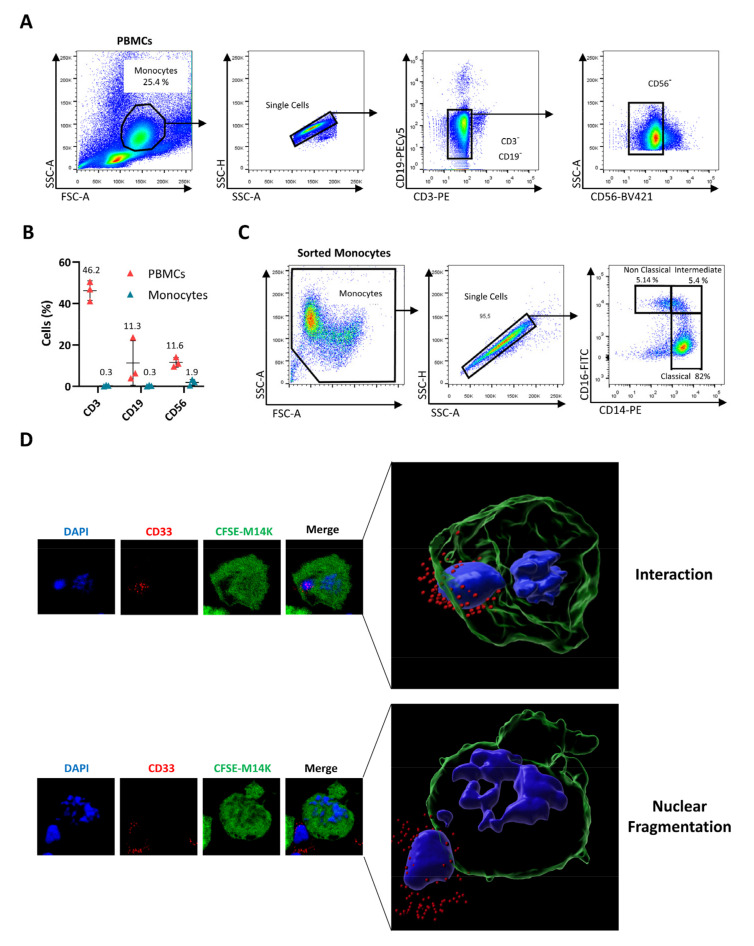
Cell-to-cell contact between blood-derived monocytes and M14K cells. (**A**) PBMCs were isolated from buffy coats of healthy donors by Ficoll gradient centrifugation. After labeling with CD3-PE, CD19-PECy5, and CD56-BV421 antibodies, monocytes were negatively sorted by flow cytometry. (**B**) Flow cytometry analysis of PBMCs and sorted monocytes labeled with CD3-PE, CD19-PECy5, and CD56-BV421 antibodies. The graph represents the percentages of labeled events before (PBMCs) and after cell sorting (Sorted Monocytes). (**C**) Isolated monocytes were labeled with anti-CD14-PE and anti-CD16-FITC. Representative plots are shown. (**D**) M14K cells were labeled with CFSE, incubated for 4 h, and then cocultured with freshly isolated monocytes at a ratio of 4/1. After 24 h of culture on coverslips, cells were labeled with an anti-CD33 primary antibody followed by an Alexa Fluor 647 conjugate. Cells were further fixed with PFA 4%, stained with DAPI, and analyzed with a HR confocal microscope. Images were computed with Imaris. Representative interactions between CFSE-positive M14K cells (in green) and CD33-labeled monocytes (in red) are shown. Nuclei fragmentation is revealed by DAPI staining (in blue).

**Figure 3 cancers-14-02164-f003:**
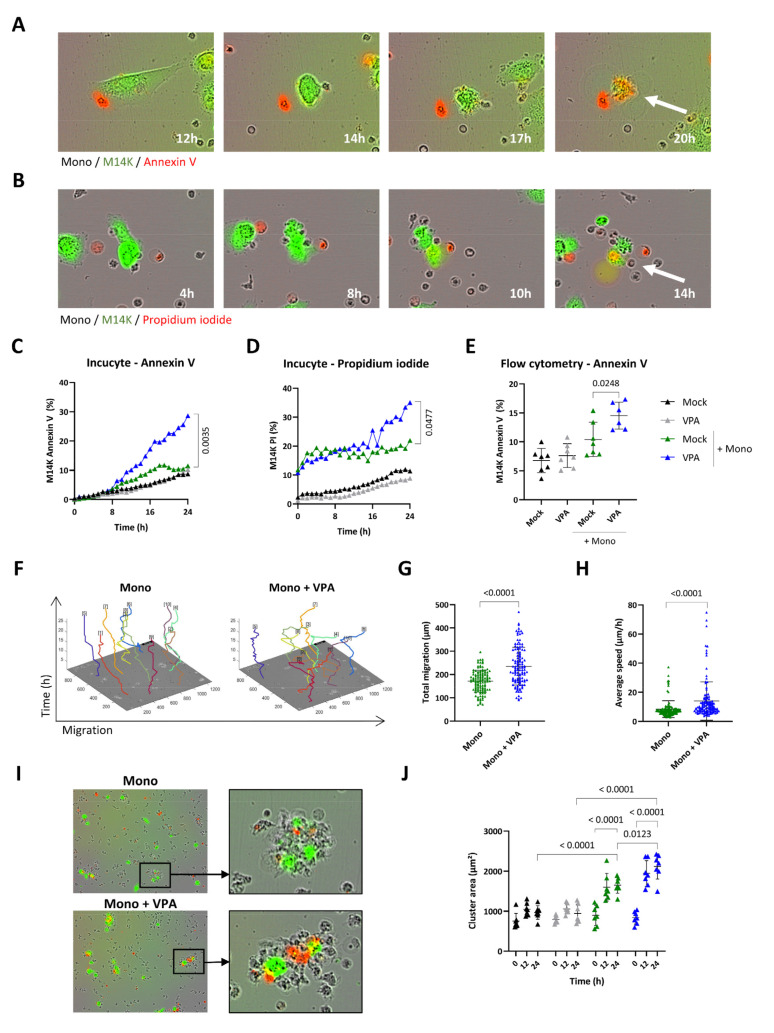
Effect of VPA on primary monocyte migration and cytotoxicity. (**A**,**B**) Blood-derived monocytes (Mock) were cocultured with CFSE-labeled M14K cells at a 20/1 ratio. Representative Incucyte kinetics upon labeling with annexin V-APC (**A**) or propidium iodide (**B**). (**C**,**D**) Single-cell imaging data from nine independent experiments were recorded by Incucyte. The graphs represent the mean percentages of CFSE-labeled M14K cells staining for annexin V (**C**) or propidium iodide (**D**). Statistical significance was evaluated using a one-way ANOVA with Tukey’s multiple comparison test. (**E**) After 24 h of coculture, cells were stained with annexin V-APC and analyzed by flow cytometry. Mean (±standard deviation) percentages of CFSE-positive M14K cells labeled with annexin V were deduced from seven independent experiments. Statistical significance was evaluated using one-way ANOVA with Tukey’s multiple comparison test. (**F**) Based on Incucyte images, cell motility of 10 monocytes per condition performed in triplicate was determined with the CellTracker software. Total migration (**G**) and average speed (**H**) data are the means (±standard deviations) of four independent experiments. The statistical significance was evaluated with non-parametric Kruskal–Wallis followed by a Dunn’s multiple comparison test. (**I**) Representative Incucyte images of clusters formed by monocytes and CFSE-labeled M14K cells in the presence of mock or VPA. (**J**) Using ImageJ, the mean area of 10 clusters formed by M14K cells alone or in interaction with one or several monocytes was measured in triplicate at 0, 12, and 24 h. Data are the means (±standard deviations) of seven independent experiments. The statistical significance was evaluated with a two-way ANOVA followed by a Tukey’s multiple comparison test.

**Figure 4 cancers-14-02164-f004:**
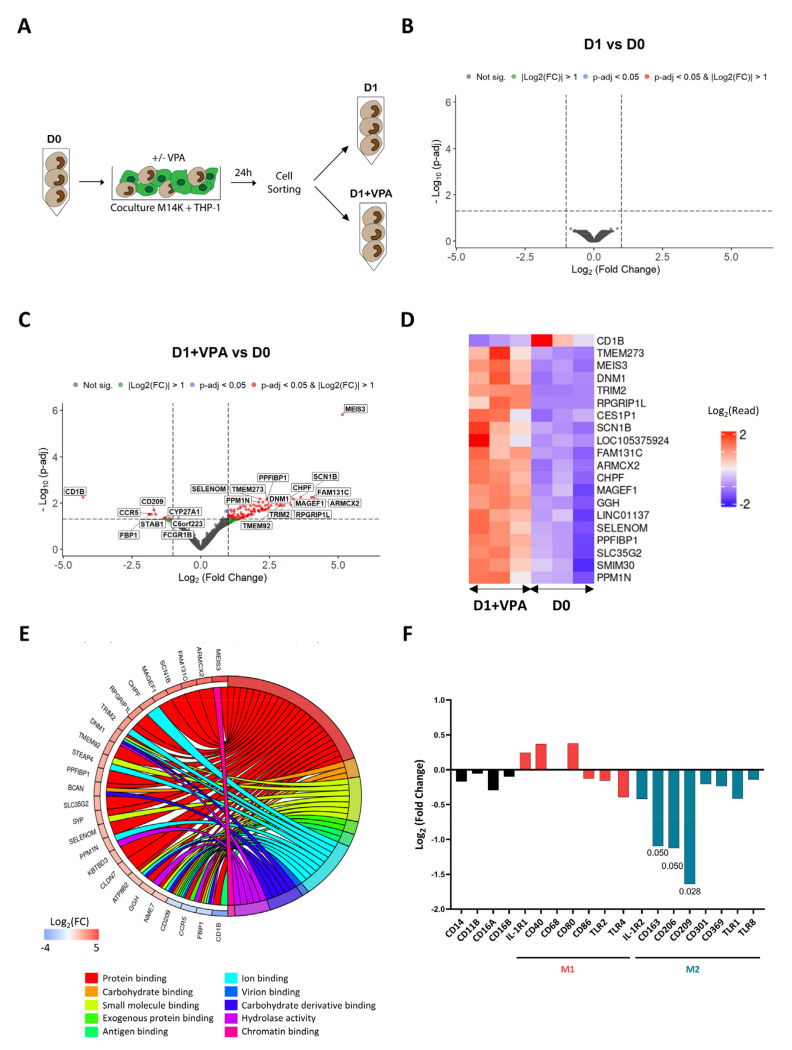
Transcriptomic profile of VPA-treated monocytes after cocultivation with MPM cells. (**A**) Experimental design. Blood-derived monocytes (D0) were cocultured with CFSE-labeled M14K cells in the absence (D1) or presence of VPA (2 mM) (D1 + VPA) for 24 h. After coculture, monocytes were isolated using a flow cytometry cell sorter. The experiment was performed independently with monocytes isolated from 3 different healthy donors. After extraction of ribonucleic acids, mRNA was purified, converted to cDNA libraries, and sequenced with a Novaseq 6000 device. (**B**) Volcano plot of differentially expressed genes between D1 and D0 conditions with Log_2_(Fold Change) threshold: 1, p-adj threshold: 0.05, shrinkage method: ashr. (**C**) Volcano plot of differentially expressed genes between D1 + VPA and D0 conditions with Log_2_(Fold Change) threshold: 1 and p-adj threshold: 0.05, shrinkage method: ashr. (**D**) Heatmap of the most significant DEGs deduced from 3 independent donors in the conditions D1 + VPA and D0. The color key represents the Log_2_(read count) (DEGs) from low (blue) to high (red). (**E**) Representative chord diagram of the 10 most significant pathways from GO: Molecular Functions (GO:MF) between the conditions D1 + VPA and D0. The chords show a detailed relationship between Log_2_(Fold Change) of DEGs (left semicircle) and their enriched GO:MF pathways (right semicircle). The color key represents the Log_2_(Fold Change) from low (blue) to high (red). The GO enrichment analysis was performed by considering all significant genes (p-adj < 0.05). (**F**) Log_2_(Fold Change) of selected membrane receptors between the conditions D1 + VPA and D0.

## Data Availability

Publicly available datasets analyzed in this study can be found on the library: https://www.ncbi.nlm.nih.gov/Traces/study/?acc=PRJNA807510&o=acc_s%3Aa, accessed on 21 February 2022.

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
