# Peer review of "HDAC Inhibition with Valproate Improves Direct Cytotoxicity of Monocytes against Mesothelioma Tumor Cells"

_cancers, 2022, doi:10.3390/cancers14092164_

Round 1
Reviewer 1 Report
Dear Author,
it is an interesting work exploring the effects of the histone acetylation inhibition by valproic acid (VPA) and methyltransferase inhibition by an EZH2 inhibitor (EPZ) on the cytotoxic activity of human blood-derived monocytes on mesothelioma cells. It is also a well-written work with interesting conclusions supported by the results.
I recommend to address the following issues before publication:
In paragraph 1: "HDAC inhibition promotes THP-1 cytotoxicity to MPM cells", it is stated: "The first goal of the study was to investigate the ability of monocytes to induce cancer cell death by direct cell-to-cell contact "concluding that "Both fluorescent markers (Annexin-V and PI) labeled CFSE + M14K in contact with THP-1 monocytes, indicating the onset of cell death ". However, it is also stated that "No statistically significant difference was observed between the different conditions: absence of monocytes (mock, in black), presence of THP-1 (in green) or THP-1 differentiated with PMA (in orange)". How is it possible that THP-1 monocytes induce M14K cell death, in the same way as in mock (absence of monocytes)?
Then, the effects of VPA and EZH2 on M14K ± THP-1 were analyzed and it was concluded that VPA significantly increased THP-1-mediated cytotoxicity toward M14K cells (in blue, Figure 1D and 1E; Figure A1). In contrast, a histone methyltransferase EZH2 inhibitor of the Polycomb 2 repressive complex (EPZ) did not have statistically significant effects in eight independent experiments. Then, it was performed a similar experiment to support the hypothesis in biphasic ZL34 MPM cells. However, in this experiment, the EZH inhibitor was not tested.
Other minor remarks:
- ZL34 cells (CVCL-5906) are referred to as non-epithelial. Why didn't the authors specify their mixed phenotype?
- Please provide a catalog number of CFSE (BioChemika) and a link to the company. I can't find it on the web.
- Please correct in Figure A1 dehydrogenase with dehydrogenase
- Figure A2. Direct cytotoxicity of THP-1 in contact with mesothelioma ZL34 cells: "THP-1 cells were cultured in the presence of 100 ng / mL of PMA or 2.5 mM of VPA for 48 hours". Please use the same unit of measurement. Similarly, EPZ concentrations are sometimes indicated in µM, sometimes in µg / ml. Please standardize the units of measurement
- The numbers of the figures are wrong
Reviewer 2 Report
The authors of the manuscript "HDAC inhibition with valproate improves direct cytotoxicity of monocytes against mesothelioma tumor cells" have performed an interesting study investigating the potential use of valproate in increasing cytotoxic effects from monocytes. The study is well-structured, performed with required controls and clear presentation.
In my opinion, the manuscript is suitable for publication after making the figure legends more concise and with minor spell checks.
Reviewer 3 Report
The study proposed by Hoyos et al. is interesting and provides the rationale for the improvement of therapies for MPM treatment, that today is still a therapy orphan disease.
In my opinion, the major issue is that the majority of the experiments are performed only in one cell line (M14K). Considering the great heterogeneity of MPM, it would be advantageous to confirm these results in at least an additional cell line of a different histotype. If not, I suggest to the authors to include this limitation in the discussion, highlighting the heterogeneity of MPM.
Other minor concerns:
-
I don’t understand why in Fig. 1B-C we observe cell death upon contact between M14K and THP1 cells while in Fig. 1D-E there is no increase in apoptosis of M14K cells co-cultured with THP1 either undifferentiated or PMA-treated (in this case apoptosis is even lower) compared to M14K cells alone. However, quantification of apoptosis by flow cytometry (Fig. 1F) show an increase of apoptosis of M14K cells when co-cultured with THP1 PMA-treated. How do you explain these non-concordant results?
-
Again in Fig. 1B-C it is not clear whether THP1 cells used for co-culture are untreated or PMA-treated. If the latter one, why the difference with figure 1I middle panel? (In Fig. 1I THP1 cells show positivity to Annexin V, while not in Fig. 1B).
-
Migration and speed of monocytes are not increased using THP1 cells (Fig. 1 K-L) while they are increased using monocytes derived from primary cultures (Fig. 3 G-H). How do authors explain this discrepancy?
-
Figures nomenclature is incorrect in the figure legend: Figure 2 is called figure 1 and so on for the following ones.
